# GPS: A PROBABILISTIC DISTRIBUTIONAL SIMILARITY WITH GUMBEL PRIORS FOR SET-TO-SET MATCHING

**Ziming Zhang**[1],[*] **Fangzhou Lin**[1],[†] **Haotian Liu**[1],[‡] **Jose Morales**[1], **Haichong Zhang**[1],
**Kazunori Yamada**[2], **Vijaya B Kolachalama**[3], **Venkatesh Saligrama**[3]

[1]Worcester Polytechnic Institute, USA  [2]Tohoku University, Japan  [3]Boston University, USA
`{zzhang15, flin2, hliu8, jamorales, hzhang10}@wpi.edu`
`yamada@tohoku.ac.jp, {vkola, srv}@bu.edu`

## ABSTRACT

Set-to-set matching aims to identify correspondences between two sets of unordered items by minimizing a distance metric or maximizing a similarity measure. Traditional metrics, such as Chamfer Distance (CD) and Earth Mover's Distance (EMD), are widely used for this purpose but often suffer from limitations like suboptimal performance in terms of accuracy and robustness, or high computational costs - or both. In this paper, we propose an effective set-to-set matching similarity measure, *GPS*, based on Gumbel prior distributions. These distributions are typically used to model the extrema of samples drawn from various distributions. Our approach is motivated by the observation that the distributions of minimum distances from CD, as encountered in real-world applications such as point cloud completion, can be accurately modeled using Gumbel distributions. We validate our method on tasks like few-shot image classification and 3D point cloud completion, demonstrating significant improvements over state-of-the-art loss functions across several benchmark datasets. Our demo code is publicly available at `https://github.com/Zhang-VISLab/ICLR2025-GPS`.

## 1 INTRODUCTION

**Problem.** Set-to-set matching involves comparing and identifying correspondences between two sets of items, which can be modeled as a bipartite graph matching problem. In this framework, the items in each set are represented as nodes on opposite sides of a bipartite graph, with the edges representing the correspondences. This task is challenging due to several factors: (1) The matching process must be invariant to the order of both the sets and the items within them; (2) Finding effective representations for each set is critical, as it greatly influences performance; (3) Computing a meaningful similarity score between sets, often used as a loss function for learning feature extractors, is a nontrivial problem. Our goal is to develop a similarity measure that is both effective and efficient for set-to-set matching.

**Distance Metrics.** Set-to-set matching has been extensively studied in various research fields (Chang et al., 2007; Zhou et al., 2017a; Saito et al., 2020; Jurewicz and Derczynski, 2021; Yu et al., 2021a; Kimura et al., 2023). Several distance metrics, such as Chamfer Distance (CD) (Yang et al., 2020), Earth Mover's Distance (EMD) (Zhang et al., 2020a; Yang et al., 2024), and Wasserstein Distance (WD) (Zhu and Koniusz, 2022), are commonly used to evaluate set-matching scores in different applications. These metrics can generally be considered special cases of optimal transportation (OT) for graph matching (Saad-Eldin et al., 2021), which aims to find the most efficient way to move a distribution of "materials" to another distribution of "consumers" by minimizing the total transportation cost. However, it is well known that these distance metrics often face challenges such as poor performance (*e.g.,* matching accuracy and robustness of CD (Lin et al., 2023a)) or high computational complexity (*e.g.,* EMD and WD (Nguyen et al., 2021; Rowland et al., 2019; Kolouri

---

[*]First co-author, project leader
[†]First co-author, responsible for experiments on point cloud completion
[‡]First co-author, responsible for experiments on few-shot classification

et al., 2019)), or both, which limit their applicability in large-scale learning. To address this issue, some new distance metrics, such as density-aware CD (DCD) (Wu et al., 2021), HyperCD (Lin et al., 2023b) and InfoCD (Lin et al., 2023a), have been developed as training losses in the literature to improve matching performance while achieving linear complexity similar to CD.

**Limitations of Traditional Distance Metrics.** Existing metrics fail to adequately capture the similarity between sets in terms of their underlying distributions. In many real-world applications, such as classification, the primary concern is not the direct similarity between individual data instances but rather their similarity in some (latent) space, such as class labels.

For instance, in Figure 1 in the context of Chamfer Distance (CD), the distance between the sets $\circ$ and $\times$ is larger than that between the sets $\circ$ and $+$, despite $\circ$ and $\times$ being sampled from the same underlying distribution. This highlights a critical issue: when asking *"how likely are two sets of points to come from the same distribution?"* CD and similar metrics may fail to reflect distributional similarity accurately. Consequently, traditional distance metrics are effective at measuring differences in observations (*e.g.,* point clouds or images) but may struggle to capture deeper, potentially unknown *abstractions* such as distributions or class labels.

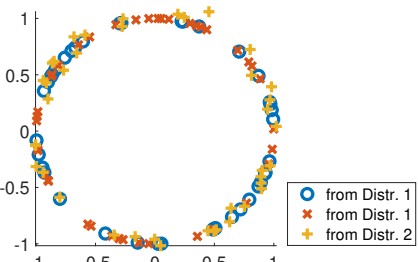

Figure 1: Illustration of three point sets randomly sampled from a circular distribution (*i.e.,* Distr. 1) and a Gaussian whose mean is conditional on the circle (*i.e.,* Distr. 2).

**Distributional Similarity.** This concept is widely applied in natural language processing to assess word similarities based on the contexts in which they appear (Lee, 2000). For example, to measure the similarity between words $u$ and $v$, both words can be represented in a vector space by counting their co-occurrences with words from a predefined vocabulary within local contexts such as sentences or paragraphs. Various similarity functions are then applied to these vectors to compute their similarity. To adapt this idea for set-to-set matching, we treat the K nearest neighbors (KNNs) of a point from the opposite set as its "co-occurrences".

**Gumbel Distributions.** In probability theory, Gumbel distributions are used to model the distribution of extreme values (maxima or minima) from various samples. We find that Gumbel distributions effectively model the KNN distances between two sets of feature vectors. This is shown in Figure 2, where the two 2D point sets are sampled from the circular distribution shown in Figure 1. We compute the negative-log distances between each point in one set and its 1st, 2nd, and 3rd nearest neighbors in the other set, transforming these distances into normalized histograms representing probabilities. These probability distributions are then fitted with Gumbel distributions (see Definition 1).

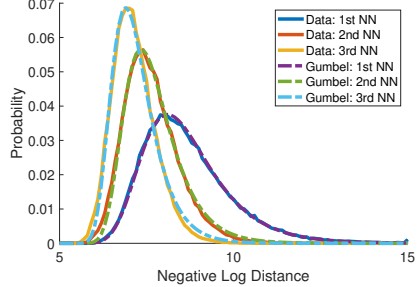

Figure 2: Data fitting with Gumbel distributions for 1st, 2nd, 3rd smallest distance distributions between two 2D point sets.

**Our Approach and Contributions.** We propose a novel probabilistic method for set-to-set matching called GPS (Gumbel Prior Similarity) based on distributional similarity and Gumbel distributions. This method measures the similarity between the underlying distributions generating the sets. Specifically, we use the log-likelihood of Gumbel distributions as our similarity measure, modeling the distributions of negative-log distances between the KNNs of the sets. GPS can be seamlessly integrated into existing neural network training frameworks, for instance, by using the negative of GPS as a loss function, while maintaining the same linear computational complexity as CD. We propose a comprehensive analysis of GPS and its influence on learning behavior. We are the first to leverage statistical information from KNNs for set-to-set matching. To demonstrate the efficacy and efficiency of GPS, we conduct extensive experiments on tasks such as few-shot image classification and 3D point cloud completion, achieving state-of-the-art performance across several benchmark datasets.

## 2 RELATED WORK

**Set-to-Set Matching.** Many tasks in computer vision and machine learning, such as multiple instance learning (Ilse et al., 2018; Maron and Lozano-Pérez, 1997), shape recognition (Su et al., 2015; Shi et al., 2015), and few-shot image classification (Afrasiyabi et al., 2022), can be framed as set-to-set matching problems, where the goal is to identify correspondences between sets of instances. The unordered nature of these sets requires the extraction of invariant features that are not affected by the sequence of elements (Choi et al., 2018). Some research addresses this challenge by modifying neural network architectures. For example, Vinyals et al. (2016) introduced matching networks for one-shot learning, while Lee et al. (2019) proposed the Set Transformer, which uses an attention-based mechanism to model interactions among elements in input sets. Saito et al. (2020) developed exchangeable deep neural networks. Recently, Kimura (2022) analyzed the generalization bounds for set-to-set matching with neural networks. The loss function for training these networks must also maintain order-invariance to effectively calculate distance functions between pairs of instances within the sets, as seen in DeepEMD (Zhang et al., 2020a).

**Similarity (Metric) Learning.** This research focuses on developing functions that measure the correlation between two objects (Ma and Manjunath, 1996; Balcan and Blum, 2006; Yu et al., 2008), and has been successfully applied to various applications, including face recognition (Faraki et al., 2021; Cao et al., 2013), few-shot image classification (Zhang et al., 2020a; Oh et al., 2022), emotion matching (Lin et al., 2016), and re-identification (Zhou et al., 2017a; Liao et al., 2017). Recently, these techniques have been integrated into deep learning (Liu et al., 2019; Cheng et al., 2018; Ma et al., 2021; Liao et al., 2017; Zhou et al., 2017b) for representation learning in embedding spaces, where objects from the same set are closer together, and objects from different sets are further apart. However, learning a model from all sample pairs is challenging due to high computational complexity and poor local minima during training (Kaya and Bilge, 2019; Qian et al., 2019; Huang et al., 2016). Thus, designing effective and efficient loss functions is a key issue in deep similarity learning (Elezi et al., 2020; Wang et al., 2019). In contrast, we address this problem by utilizing the statistics of minimum distances between the items of sets to improve computational efficiency and performance.

**Few-Shot Classification.** This task aims to train a classifier to recognize both seen and unseen classes with limited labeled examples (Chen et al., 2019a). During training, the model learns a generalized classification ability in varying classes (Oreshkin et al., 2018; Finn et al., 2017). In the testing phase, when presented with entirely new classes, the model classifies by calculating the closest similarity measurement (Chen et al., 2019a; Naik and Mammone, 1992). Formally, in few-shot learning, the training set includes many classes, each with multiple samples (Li et al., 2018; Ren et al., 2018). For example, $C$ classes are randomly selected from the training set and $K$ samples from each category (totaling $C \times K$ samples) are used as input to the model. A batch of samples from the remaining data in these $C$ classes is then used as the model's prediction target (batch set). The model must learn to distinguish these $C$ classes from $C \times K$ pieces of data, a task known as a $C$-way $K$-shot problem.

**Point Cloud Completion.** This task involves an important objective of inferring the complete shape of an object or scene from incomplete raw point clouds. Recently, numerous deep learning approaches have been developed to address this problem. For example, PCN (Yuan et al., 2018) extracts global features directly from point clouds and generates points using the folding operations from FoldingNet (Yang et al., 2018). Zhang et al. (2020b) proposed extracting multiscale features from different network layers to capture local structures and improve performance. Attention mechanisms, such as the Transformer (Vaswani et al., 2017), excel in capturing long-term interactions. Consequently, SnowflakeNet (Xiang et al., 2021), PointTr (Yu et al., 2021b), and SeedFormer (Zhou et al., 2022) emphasize the decoder component by incorporating Transformer designs. PointAttN (Wang et al., 2022) is built entirely on Transformer foundations. Recently, Lin et al. (2023b) introduced a HyperCD loss for training neural networks that defines traditional CD in a hyperbolic space. Furthermore, Lin et al. (2023a) proposed an InfoCD loss by incorporating the contrastive concept into the CD formula.

**Gumbel Distribution.** In machine learning and computer vision, the Gumbel distribution has been widely used in sampling methods (Maddison et al., 2014; Kool et al., 2019) and reparameterization techniques (Huijben et al., 2022; Kusner and Hernández-Lobato, 2016; Potapczynski et al., 2020). For example, Hancock and Khoshgoftaar (2020) introduced the Gumbel-Softmax reparameterization to enable differentiable sampling from a discrete distribution during backpropagation in neural networks.

## 3 APPROACH

### 3.1 PRELIMINARIES

**Notations & Problem Definition.** We denote $\mathcal{X}_1 = \{x_{1,i}\} \sim \mathcal{P}_1, \mathcal{X}_2 = \{x_{2,j}\} \sim \mathcal{P}_2$ as two sets of points (or items) that are sampled from two unknown distributions $\mathcal{P}_1, \mathcal{P}_2$, respectively, and $K$ as the number of nearest neighbors considered for each point. Also, we refer to $|\cdot|$ as the cardinality of a set, and $\|\cdot\|$ as the $\ell_2$-norm of a vector. Given these notations, our goal is to predict the set-to-set similarity, $\kappa(\mathcal{X}_1, \mathcal{X}_2)$, based on the conditional probability $p(\mathcal{P}_1 = \mathcal{P}_2 | \mathcal{X}_1, \mathcal{X}_2)$.

**Gumbel Distributions.** Recall that the Gumbel distribution is used to model the distribution of the *maximum* (or minimum by replacing the maximum with the negative of minimum) of a number of samples from various distributions. Here we list the definition of a Gumbel distribution as follows:

**Definition 1** (Gumbel Distribution). *The probability density function (PDF) of a Gumbel distribution with parameters $\mu \in \mathbb{R}, \sigma > 0$, denoted as $Gumbel(\mu, \sigma)$, for a random variable $x \in \mathbb{R}$ is defined as*

$$p(x) = \frac{1}{\sigma} \exp\{-(y + \exp\{-y\})\}, \ where \ y = \frac{x - \mu}{\sigma}. \tag{1}$$

### 3.2 GPS: GUMBEL PRIOR SIMILARITY FOR SET-TO-SET MATCHING

**Distributional Signatures.** Given two sets of points $\mathcal{X}_1, \mathcal{X}_2$, we define the set of Euclidean distances from each point in one set to its KNNs in the other set as their distributional signature:

$$\mathcal{D}(\mathcal{X}_1, \mathcal{X}_2) = \left\{ d_{min}^{(k)}(x_{1,i}) = \|x_{1,i} - x_{2,i_k}\|, d_{min}^{(k)}(x_{2,j}) = \|x_{2,j} - x_{1,j_k}\| \mid \forall k \in [K], \forall i, \forall j \right\} \tag{2}$$

where $i_k$ (*resp.* $j_k$) denotes the index of the $k$-th nearest neighbor in $\mathcal{X}_2$ (*resp.* $\mathcal{X}_1$) for $x_{1,i}$ (*resp.* $x_{2,j}$), leading to an unordered set of $K(|\mathcal{X}_1| + |\mathcal{X}_2|)$ values.

**Probabilistic Modeling.** To compute GPS, we introduce the Gumbel distributions and distributional signatures as latent variables, as shown in Figure 3, and propose a probabilistic framework as follows:

$$p(\mathcal{P}_1 = \mathcal{P}_2 \mid \mathcal{X}_1, \mathcal{X}_2) = \sum_{q \in \mathcal{Q}} \sum_{d_{min} \in \mathcal{D}} p(\mathcal{P}_1 = \mathcal{P}_2, q, d_{min} \mid \mathcal{X}_1, \mathcal{X}_2)$$

$$= \sum_{q \in \mathcal{Q}} \sum_{d_{min} \in \mathcal{D}} p(q)p(\mathcal{P}_1 = \mathcal{P}_2 \mid q)p(d_{min} \mid q, \mathcal{X}_1, \mathcal{X}_2), \tag{3}$$

where $q \in \mathcal{Q}$ denotes a Gumbel distribution. Figure 3 illustrates the probability decomposition where a Gumbel distribution $q$ is selected for measuring set-to-set similarity based on the distributional signatures between sets. Note that the distributions of each $k$-th nearest neighbor distances could be modeled using a mixture of $M$ independent Gumbel distributions. Consequently, Equation (3) can be further rewritten as follows:

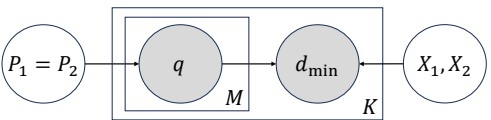

Figure 3: Graphical model for computing the conditional probability $p(\mathcal{P}_1 = \mathcal{P}_2 \mid \mathcal{X}_1, \mathcal{X}_2)$.

$$p(\mathcal{P}_1 = \mathcal{P}_2 \mid \mathcal{X}_1, \mathcal{X}_2) \propto \sum_{k,m} \left[ \sum_i p\left(d_{min}^{(k)}(x_{1,i}); \alpha_{k,m}, \beta_{k,m}\right) + \sum_j p\left(d_{min}^{(k)}(x_{2,j}); \alpha_{k,m}, \beta_{k,m}\right) \right], \tag{4}$$

where $k \in [K], m \in [M]$ and $\{\alpha_{k,m}, \beta_{k,m}\}$ denote the predefined Gumbel parameters of the $m$-th mixture for fitting the $k$-th nearest neighbor distances. For simplicity, we use the same mixture for both sets (but they can be different). Note that here we take $p(q)$ and $p(\mathcal{P}_1 = \mathcal{P}_2 \mid q)$ as two constants with no prior knowledge.

**Reparametrization with Minimum Euclidean Distances.** The KNN distances are sets of (conditional) minima, not maxima. Therefore, we need to convert these minimum distances to some form of maximum values that can be modeled using Gumbel. Considering both computational complexity and

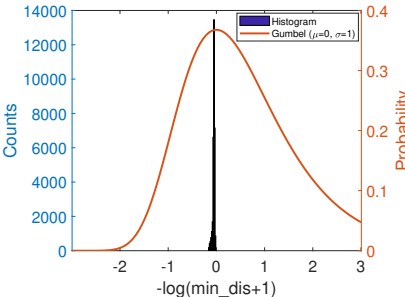 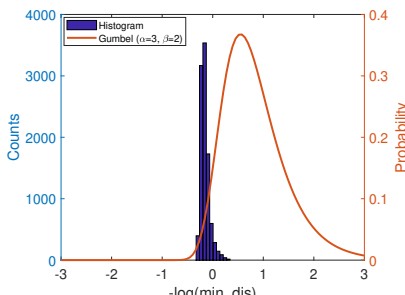

Figure 4: Illustration of the distributions of minimum distances (with the learned best model) at test time for **(left)** point cloud completion and **(right)** few-shot image classification.

learning objectives, we propose reparametrizing $x \overset{def}{=} -\log(d_{min} + \delta)$ in Equation (1), where $\delta \geq 0$ denotes a distance shift (see Section 3.3 for more discussions), and rewrite Equation (4) accordingly as follows, which formally defines our GPS:

$$\kappa(\mathcal{X}_1, \mathcal{X}_2)$$

$$\overset{def}{=} \sum_{k,m} \left[ \sum_i D_{min}^{(k,m)}(x_{1,i}) \exp\{-D_{min}^{(k,m)}(x_{1,i})\} + \sum_j D_{min}^{(k,m)}(x_{2,j}) \exp\{-D_{min}^{(k,m)}(x_{2,j})\} \right], \quad (5)$$

where $\mu_{k,m} = \frac{\log \alpha_{k,m}}{\beta_{k,m}}, \sigma_{k,m} = \frac{1}{\beta_{k,m}} > 0, \alpha_{k,m} = e^{\frac{\mu_{k,m}}{\sigma_{k,m}}} > 0, \beta_{k,m} = \frac{1}{\sigma_{k,m}} > 0, D_{min}^{(k,m)}(x_{1,i}) = \alpha_{k,m} \left[ d_{min}^{(k,m)}(x_{1,i}) + \delta_{k,m} \right]^{\beta_{k,m}}, D_{min}^{(k,m)}(x_{2,j}) = \alpha_{k,m} \left[ d_{min}^{(k,m)}(x_{2,j}) + \delta_{k,m} \right]^{\beta_{k,m}}$, respectively.

## 3.3 ANALYSIS

**Proposition 1.** *Letting $f(x) = xe^{-x}, x > 0$, then $f$ has a unique maximum at $x = 1$, and is concave when $0 < x < 2$, otherwise convex when $x > 2$,*

*Proof.* By taking the first-order derivative of $f$, $\nabla f(x) = (1-x)e^{-x}$, and setting it to 0, we can show that $f$ has a unique extremum at $x = 1$. By taking the second-order derivative, $\nabla^2 f(x) = -(2-x)e^{-x}$, we can easily show its convexity and concavity by looking at the signs of the values. □

**Gradients decrease (exponentially) when minimum distances approach the mode.** This can be easily found from the form of the gradient in the proof above. As a result, the contribution of the gradient from a data sample in backpropagation will be smaller if its minimum distance is closer to the mode, making it less important in training. In other words, the training will focus more on correcting poor predictions by adaptively assigning higher weights in backpropagation. Note that all $\alpha$'s and $\beta$'s in GPS have an impact on the gradients in learning.

**Distance Shifting.** In many applications, such as point cloud completion, the objective is to minimize the distances between predictions and the ground truth. In these scenarios, the mode of a Gumbel distribution is expected to be 0, which results in $\delta_{k,m} = (\alpha_{k,m})^{-\frac{1}{\beta_{k,m}}} > 0$. To maximize similarity during training, all minimum distances should be enforced to approach this mode. Therefore, our reparametrization includes $d_{min} + \delta_{k,m}$ because, as $d_{min}$ approaches 0, $-\log(d_{min})$ would tend toward infinity, contradicting the goal of maximizing similarity by approaching the mode. However, $-\log(d_{min} + \delta_{k,m})$ avoids this problem. Figure 4 (left) shows an example using a randomly sampled airplane instance from ShapeNet (see our experiments on point cloud completion for more details), where all minimum distances are approaching 0 as well as the Gumbel mode, achieving both goals of minimizing distance and maximizing similarity simultaneously.

**Similarity Embedding.** Also in many applications, such as few-shot image classification, the objective is to minimize some loss, *e.g.,* cross-entropy for classification, that depends on set-to-set similarity. In such cases, the minimum distances are not necessarily as small as possible, neither matching with the Gumbel distributions, as long as they help distinguish the data towards minimizing the loss, *e.g.,* higher similarities for positive pairs and lower for negative pairs. Therefore, we do

Table 1: $\alpha$ and $\beta$ search with 1st nearest neighbors for few-shot classification (%) on CIFAR-FS.

| $\alpha$ | 0.5 | 1 | 1 | 2 | 2 | 2 | 2 | 5 | 2 | 2 | 5 | 2 | 2.5 | 3 | 4 | 6 |
|---|---|---|---|---|---|---|---|---|---|---|---|---|---|---|---|---|
| $\beta$ | 2 | 0.5 | 1 | 0.5 | 2 | 1 | 1.5 | 2 | 0.5 | 3.5 | 4 | 2 | 2 | 2 | 2 | 2 |
| Acc | 72.75 | 72.51 | 73.23 | 73.54 | 73.67 | 73.14 | 73.08 | **74.22** | 72.14 | 73.23 | 73.89 | 73.71 | 73.52 | 74.14 | 74.17 | 73.80 |

Table 2: $\alpha$ and $\beta$ search using 2nd nearest neighbors performance (%) on CIFAR-FS.

| $\alpha$ | 0.1 | 0.5 | 1 | 1 | 2 | 3 | 2 | 2 | 2 | 2 | 2.5 | 3.5 | 4 | 5 |
|---|---|---|---|---|---|---|---|---|---|---|---|---|---|---|
| $\beta$ | 2 | 2 | 0.5 | 2 | 2 | 2 | 0.5 | 2.5 | 3.5 | 4 | 2 | 2 | 2 | 5 |
| Acc | 71.67 | 72.84 | 73.17 | 73.04 | 73.08 | **73.66** | 72.45 | 72.85 | 73.23 | 72.67 | 73.37 | 72.95 | 73.44 | 60.74 |

not need distance shifting and simply set $\delta = 0$. Figure 4 (right) illustrates a distribution of learned minimum distances for few-shot image classification (see our experiments for more details), compared with the Gumbel distribution used in training, where all the minimum distances are close to 1.

## 4 EXPERIMENTS

### 4.1 FEW-SHOT IMAGE CLASSIFICATION

We conduct our experiments on five benchmark datasets: *mini*ImageNet (Vinyals et al., 2016), *tiered*ImageNet (Ren et al., 2018), Fewshot-CIFAR100 (Oreshkin et al., 2018), CIFAR-FewShot (Bertinetto et al., 2018), and Caltech-UCSD Birds-200-2011 (Wah et al., 2011). We follow the standard protocols in the literature to split these datasets for our training and testing.

Table 3: Performance (%) using mixtures of Gumbels and NNs on CIFAR-FS.

| 1st NN $(\alpha, \beta)$ | (5, 2) | (3, 2) (2, 2) | (3, 2) (2, 2) | (2.5, 2) (3, 2) | (3, 2) (5, 2) |
|---|---|---|---|---|---|
| 2nd NN $(\alpha, \beta)$ | (3, 2) | (2.5, 2) (4, 2) | (4, 2) (2, 2) | (2, 2) (4, 2) | (4, 2) (1, 2) |
| Acc | 73.89 | **74.17** | 74.09 | 74.12 | 73.62 |

**Network Framework.** We take the framework in DeepEMD (Zhang et al., 2020a)[1] for fair comparisons with different losses. Following the literature (Sun et al., 2019; Chen et al., 2021; Liu et al., 2021; Chikontwe et al., 2022), we employ ResNet-12 as our network backbone. We notice that DeepEMD and its extensions have different implementations. Precisely, we use the DeepEMD-FCN network for comparisons and report our results in our experiments. We also observe that GPS with the other implementations of DeepEMD can still significantly improve the performance. For instance, by replacing FCN with Grid and Sampling layers (Zhang et al., 2022), our GPS in 1-shot 5-way can improve 1.2% over DeepEMD with Grid and 1.3% with Sampling.

**Training Objective.** Same as DeepEMD, we optimize the following objective for the few-shot tasks:

$$\min_{\theta, \omega} \sum_{u,v} \ell(\kappa(\mathcal{X}_u, \mathcal{X}_v), y_u, y_v; \omega), \text{ s.t. } \mathcal{X}_u = \{f(x_{u,i}; \theta)\}, \mathcal{X}_v = \{f(x_{v,j}; \theta)\}, \quad (6)$$

where $x_{u,i}, x_{v,j}$ stand for two patches from the $u$-th and $v$-th images with labels $y_u, y_v$, respectively, $f$ for a neural network parametrized by $\theta$, and $\ell$ for a loss parametrized by $\omega$. In our GPS, $k$ uses KNNs to determine the matches between patches, while in DeepEMD a differentiable EMD was proposed to find the matches. We reuse the framework by replacing the EMD with our GPS.

**Training Protocols.** Following DeepEMD, we re-implement all pre-training, meta-training, validation and testing stages with different loss functions, and retrain the networks from scratch. We keep all the hyperparameters the same as DeepEMD but fine-tune the loss-related hyperparameters, if exist, to report best performance. For better comparison, we rerun the public code and report our results whenever possible; otherwise, we cite the original results for the remaining methods. In each result table, the top-performing result is highlighted in bold. We conduct all experiments on a server with 10 NVIDIA RTX 6000 11G GPUs and another one with 10 NVIDIA Quadro RTX 6000 24G GPUs.

**Ablation Study.** We have conducted some experiments for few-shot image classification, including:

- *M Gumbels, K Nearest Neighbors, and Mixtures:* Table 1, Table 2 and Table 3 list our hyperparameter search results for 1st NN only, 2nd NN only, and mixtures of Gumbels and NNs, respectively. With proper tuning, the best results under different settings are close to each other, and the mixtures seem to be more robust to the data, leading to similar performance with different combinations.

---

[1]https://github.com/icoz69/DeepEMD

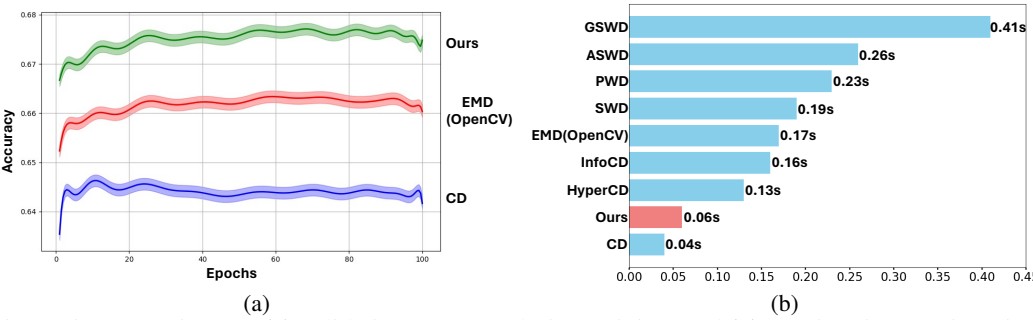

Figure 5: Comparison on **(a)** validation accuracy during training, and **(b)** running time per iteration.

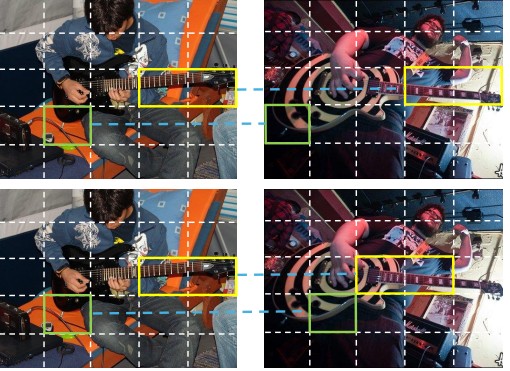

Figure 6: Visual matching results by **(top)** our GPS and **(bottom)** DeepEMD.

Table 4: Cross-domain results (%) (*mini*Imagenet → CUB) on 1-shot 5-way and 5-shot 5-way tasks.

| Model | 1-shot | 5-shot |
|---|---|---|
| ProtoNet | 50.01±0.82 | 72.02±0.67 |
| MatchingNet | 51.65±0.84 | 69.14±0.72 |
| cosine classifier | 44.17±0.78 | 69.01±0.74 |
| *linear* classifier | 50.37±0.79 | 73.30±0.69 |
| KNN | 50.84±0.81 | 71.25±0.69 |
| DeepEMD | 51.07±0.59 | 75.27±0.48 |
| CD | 51.32±0.46 | 75.61±0.54 |
| HyperCD | 51.76±0.48 | 75.98±0.42 |
| InfoCD | 51.95±0.65 | 76.23±0.39 |
| **Ours: GPS** | **52.38±0.47** | **76.59±0.52** |

- *Training Behavior & Running Time:* Figure 5 (a) shows the validation accuracy *vs.* the number of epochs, and our approach consistently outperforms the other two competitors. Figure 5 (b) shows the running time with different loss functions to train the network, where our GPS (using 1 Gumbel and 1st NN only) is the only one that can manage to preserve similar computational time to CD.
- *Visualization of Matching Results:* To understand the matching behavior, we visualize some matching results in Figure 6. Compared with DeepEMD, our approach can effectively establish correspondence between local regions more precisely with just simple nearest neighbor search.

**Comparisons with Different Losses.** Table 4, Table 5, and Table 6 summarize our comparison results on few-shot image classification on different datasets with $\alpha = 5, \beta = 2$ and 1 NN only. As we see, our approach outperforms all the competitors on all the datasets, and is more robust than the others, in general, which demonstrates the effectiveness and efficiency of GPS.

## 4.2 POINT CLOUD COMPLETION

**Datasets & Backbone Networks.** We conduct our experiments on the **five** benchmark datasets: PCN (Yuan et al., 2018), ShapeNet-55/34 (Yu et al., 2021b), ShapeNet-Part (Yi et al., 2016), and KITTI (Geiger et al., 2012). We compare our method using **seven** different existing backbone networks, *i.e.,* FoldingNet (Yang et al., 2018), PMP-Net (Wen et al., 2021), PoinTr (Yu et al., 2021b), SnowflakeNet (Xiang et al., 2021), CP-Net (Lin et al., 2022), PointAttN (Wang et al., 2022) and SeedFormer (Zhou et al., 2022), by replacing the CD loss with our GPS wherever it occurs.

**Training Objective.** Similar to InfoCD (Lin et al., 2023a), we optimize the following objective:

$$\max_{\theta} \sum_{u} \kappa(f(\mathcal{X}_u; \theta), \mathcal{Y}_u), \tag{7}$$

where $(\mathcal{X}_u, \mathcal{Y}_u)$ denotes a pair of a partial point cloud and its ground-truth complete point cloud and $f$ is a network (*e.g.,* one of the seven backbone networks) parametrized by $\theta$.

**Training Protocols.** We use the same training hyperparameters such as learning rates, batch sizes and balance factors as the original losses for fair comparisons. We conduct our experiments on a

Table 5: Results of 5-way (%) on *mini*ImageNet and *tiered*ImageNet datasets.

| Method | *mini*ImageNet | | *tiered*ImageNet | |
|---|---|---|---|---|
| | 1-shot | 5-shot | 1-shot | 5-shot |
| cosine classifier (Chen et al., 2019b) | 55.43±0.31 | 71.18±0.61 | 61.49±0.91 | 82.37±0.67 |
| ProtoNet (Snell et al., 2017) | 60.37±0.83 | 78.02±0.57 | 65.65±0.92 | 83.40±0.65 |
| MatchingNet (Vinyals et al., 2016) | 63.08±0.80 | 75.99±0.60 | 68.50±0.92 | 80.60±0.71 |
| DeepEMD (Zhang et al., 2020a) | 63.36±0.75 | 79.15±0.66 | 70.48±0.78 | 83.89±0.67 |
| CD | 63.40±0.46 | 79.54±0.39 | 70.23±0.64 | 84.01±0.31 |
| PWD (Rowland et al., 2019) | 63.92±0.77 | 78.77±0.37 | 70.69±0.92 | 83.88±0.34 |
| SWD (Kolouri et al., 2019) | 63.15±0.76 | 78.46±0.41 | 69.72±0.93 | 83.02±0.33 |
| GSWD (Kolouri et al., 2019) | 63.66±0.72 | 78.92±0.47 | 70.25±0.86 | 83.62±0.31 |
| ASWD (Nguyen et al., 2021) | 63.16±0.75 | 78.87±0.45 | 69.30±0.91 | 83.71±0.38 |
| HyperCD (Lin et al., 2023b) | 63.63±0.65 | 79.78±0.73 | 70.58±0.81 | 84.27±0.48 |
| InfoCD (Lin et al., 2023a) | 64.01±0.32 | 80.87±0.64 | 70.97±0.59 | 84.54±0.36 |
| **Ours: GPS** | **66.27±0.37** | **81.19±0.47** | **73.16±0.43** | **85.52±0.48** |

Table 6: 1-shot 5-way and 5-shot 5-way performance comparison (%).

| Method | CIFAR-FS | | Fewshot-CIFAR100 | | Caltech-UCSD Birds | |
|---|---|---|---|---|---|---|
| | 1-shot | 5-shot | 1-shot | 5-shot | 1-shot | 5-shot |
| ProtoNet | - | - | 41.54±0.76 | 57.08±0.76 | 66.09±0.92 | 82.50±0.58 |
| DeepEMD | 71.16±0.42 | 85.12±0.29 | 43.93±0.82 | 61.01±0.41 | 67.04±0.31 | 85.12±0.76 |
| CD | 71.75±0.55 | 85.48±0.51 | 44.15±0.46 | 61.22±0.58 | 67.11±0.46 | 85.31±0.59 |
| PWD | 71.58±0.31 | 84.76± 0.33 | 43.47±0.29 | 60.41±0.41 | 65.60±0.33 | 84.87±0.45 |
| SWD | 70.99±0.34 | 84.52±0.37 | 43.32±0.28 | 60.31±0.31 | 65.44±0.32 | 84.38±0.41 |
| ASWD | 71.45±0.32 | 85.26±0.35 | 43.83±0.29 | 60.89±0.34 | 65.45±0.34 | 84.76±0.39 |
| HyperCD | 72.02±0.41 | 85.77±0.43 | 44.42±0.33 | 61.59±0.74 | 67.45±0.38 | 85.42±0.75 |
| InfoCD | 72.31±0.23 | 85.91±0.39 | 44.81±0.43 | 61.87±0.66 | 67.82±0.42 | 85.90±0.68 |
| **Ours: GPS** | **74.22±0.22** | **86.98±0.23** | **46.75±0.28** | **62.91±0.47** | **69.42±0.33** | **86.78±0.62** |

server with 4 NVIDIA A100 80G GPUs and one with 10 NVIDIA Quadro RTX 6000 24G GPUs due to the large model sizes of some baseline networks.

**Evaluation.** Following the literature, we evaluate the best performance of all the methods using vanilla CD (lower is better). We also use F1-Score@1% (Tatarchenko et al., 2019) (higher is better) to evaluate the performance on ShapeNet-55/34. For KITTI, we utilize the metrics of Fidelity and Maximum Mean Discrepancy (MMD) for each method (lower is better for both metrics).

**Grid Search on $(\alpha, \beta)$.** Table 7 summarizes our search results. As we discussed in Section 3.3, all the settings produce similar performance. Considering the running time and hyperparameter search results in few-shot classification, in our experiments on point cloud completion, we only use 1st NNs and a single Gumbel distribution to fit the data, without further tuning.

Table 9: Performance comparison on ShapeNet-Part.

| Losses | L2-CD$\times 10^3$ |
|---|---|
| L1-CD | 4.16±0.028 |
| L2-CD | 4.82±0.117 |
| DCD | 5.74±0.049 |
| PWD | 14.39±0.024 |
| SWD | 6.28±0.073 |
| GSWD | 6.26±0.034 |
| ASWD | 7.52±0.058 |
| HyperCD | 4.03±0.007 |
| InfoCD | 4.01±0.004 |
| **GPS** | **3.94±0.003** |

**State-of-the-art Result Comparisons.** We summarize our comparisons as follows:

- *ShapeNet-Part:* We begin by presenting our performance comparison in Table 9 on the ShapeNet-Part dataset using CP-Net. Evidently, our approach outperforms all other competitors.
- *ShapeNet-34:* We evaluate performance across 34 seen categories (used during training) and 21 unseen categories (not used during training), detailing our results in Table 8. It is evident that our approach consistently improves the performance of baseline models, indicating its high generalizability for point cloud completion tasks.
- *ShapeNet-55:* We evaluate the adaptability of our method across datasets for tasks with greater diversity. Table 11 lists the L2-CD across three difficulty levels, along with the average. Following the literature, we present results for five categories (Table, Chair, Plane, Car, and Sofa) with over 2,500 training samples each, as shown in the table. Additionally, we provide results using the

Table 7: $\alpha$ and $\beta$ search with 1st nearest neighbors on ShapeNet-Part using CP-Net.

| $\alpha$ | 1 | 1 | 1 | 1 | 1 | 1 | 0.5 | 2 | 4 | 2 |
|---|---|---|---|---|---|---|---|---|---|---|
| $\beta$ | 0.4 | 0.8 | 1 | 1.2 | 1.6 | 2 | 1 | 1 | 1 | 2 |
| CD | 4.29 | 3.98 | **3.94** | 4.04 | 4.34 | 4.68 | 4.11 | 4.05 | 4.13 | 4.72 |

Table 8: Results on ShapeNet-34 using L2-CD$\times 1000$ and F1 score.

| Methods | 34 seen categories | | | | | 21 unseen categories | | | | |
|---|---|---|---|---|---|---|---|---|---|---|
| | CD-S | CD-M | CD-H | Avg. | F1 | CD-S | CD-M | CD-H | Avg. | F1 |
| FoldingNet | 1.86 | 1.81 | 3.38 | 2.35 | 0.139 | 2.76 | 2.74 | 5.36 | 3.62 | 0.095 |
| HyperCD + FoldingNet | 1.71 | 1.69 | 3.23 | 2.21 | 0.148 | 2.55 | 2.59 | 5.19 | 3.44 | 0.122 |
| InfoCD + FoldingNet | 1.54 | 1.60 | 3.10 | 2.08 | 0.177 | 2.42 | 2.49 | 5.01 | 3.31 | 0.157 |
| **Ours: GPS + FoldingNet** | **1.49** | **1.55** | **3.02** | **2.02** | **0.182** | **2.39** | **2.43** | **5.00** | **3.27** | **0.159** |
| PoinTr | 0.76 | 1.05 | 1.88 | 1.23 | 0.421 | 1.04 | 1.67 | 3.44 | 2.05 | 0.384 |
| HyperCD + PoinTr | 0.72 | 1.01 | 1.85 | 1.19 | 0.428 | 1.01 | 1.63 | 3.40 | 2.01 | 0.389 |
| InfoCD + PoinTr | 0.47 | 0.69 | 1.35 | 0.84 | 0.529 | 0.61 | 1.06 | 2.55 | 1.41 | 0.493 |
| **Ours: GPS + PoinTr** | **0.43** | **0.64** | **1.27** | **0.78** | **0.533** | **0.60** | **1.04** | **2.52** | **1.38** | **0.495** |
| SeedFormer | 0.48 | 0.70 | 1.30 | 0.83 | 0.452 | 0.61 | 1.08 | 2.37 | 1.35 | 0.402 |
| HyperCD + SeedFormer | 0.46 | 0.67 | 1.24 | 0.79 | 0.459 | 0.58 | 1.03 | 2.24 | 1.31 | 0.428 |
| InfoCD + SeedFormer | 0.43 | 0.63 | 1.21 | 0.75 | 0.581 | 0.54 | 1.01 | 2.18 | 1.24 | 0.449 |
| **Ours: GPS+ SeedFormer** | **0.42** | **0.61** | **1.20** | **0.74** | **0.582** | **0.52** | **1.00** | **2.15** | **1.22** | **0.451** |

F1 metric. Once again, our approach significantly improves the baseline models, especially with simpler networks. We also include some visualization results in Figure 7.

- *KITTI:* To validate the effectiveness of GPS for point cloud completion on a large-scale real-world benchmark, we follow (Xie et al., 2020) to finetune two baseline models with GPS on ShapeNetCars (Yuan et al., 2018) and evaluate their performance on KITTI. We report the Fidelity and MMD metrics in Table 10, observing that GPS consistently improves the baseline models.
- *PCN:* We also summarize the results on another benchmark dataset, PCN, with additional backbones in Table 12, showcasing state-of-the-art performance in point cloud completion.

Table 10: Results on KITTI in terms of the fidelity and MMD metrics.

| | FoldingNet | HyperCD+F. | InfoCD+F. | **GPS+F.** | PoinTr | HyperCD+P. | InfoCD+P. | **GPS+P.** |
|---|---|---|---|---|---|---|---|---|
| Fidelity $\downarrow$ | 7.467 | 2.214 | 1.944 | **1.883** | 0.000 | 0.000 | 0.000 | 0.000 |
| MMD $\downarrow$ | 0.537 | 0.386 | 0.333 | **0.302** | 0.526 | 0.507 | 0.502 | **0.449** |

## 5 CONCLUSION

In this paper, we present *GPS*, a novel, effective, and efficient similarity learning framework for set-to-set matching. Our method fits a predefined Gumbel distribution to the negative log minimum distances between set items. Originating from a probabilistic graphical model, our approach allows multiple Gumbel distributions to model the distributions of KNN distances as distributional signatures. We demonstrate superior performance in few-shot image classification and point cloud completion compared to traditional distance metrics, while maintaining a similar running time to CD.

**Limitations.** Finding optimal hyperparameters for Gumbel distributions to fit the distance distributions can be time-consuming, particularly when dealing with large sets, such as 16,384 points per object in PCN and ShapeNet-55/34. In future work, we plan to develop general guidelines for efficiently determining these hyperparameters.

## ACKNOWLEDGMENT

Vijaya B. Kolachalama is supported by grants from the National Institute on Aging's Artificial Intelligence and Technology Collaboratories (P30-AG073104 & P30-AG073105), the American Heart Association (20SFRN35460031), and the National Institutes of Health (R01-HL159620, R01-AG062109, & R01-AG083735). Venkatesh Saligrama was supported by the Army Research Office Grant W911NF2110246, AFRLGrant FA8650-22-C1039, the National Science Foundation grants CCF-2007350, CCF-1955981 and CPS-2317079.

Table 11: Results on ShapeNet-55 using L2-CD×1000 and F1 score.

| Methods | Table | Chair | Plane | Car | Sofa | CD-S | CD-M | CD-H | Avg. | F1 |
|---|---|---|---|---|---|---|---|---|---|---|
| FoldingNet | 2.53 | 2.81 | 1.43 | 1.98 | 2.48 | 2.67 | 2.66 | 4.05 | 3.12 | 0.082 |
| HyperCD + FoldingNet | 2.35 | 2.62 | 1.25 | 1.76 | 2.31 | 2.43 | 2.45 | 3.88 | 2.92 | 0.109 |
| InfoCD + FoldingNet | 2.14 | 2.37 | 1.03 | 1.55 | 2.04 | 2.17 | 2.50 | 3.46 | 2.71 | 0.137 |
| **Ours: GPS + FoldingNet** | **2.07** | **2.30** | **1.02** | **1.47** | **2.01** | **2.13** | **2.44** | **3.37** | **2.64** | **0.143** |
| PoinTr | 0.81 | 0.95 | 0.44 | 0.91 | 0.79 | 0.58 | 0.88 | 1.79 | 1.09 | 0.464 |
| HyperCD + PoinTr | 0.79 | 0.92 | 0.41 | 0.90 | 0.76 | 0.54 | 0.85 | 1.73 | 1.04 | 0.499 |
| InfoCD + PoinTr | 0.69 | 0.83 | 0.33 | 0.80 | 0.67 | 0.47 | 0.73 | 1.50 | 0.90 | 0.524 |
| **Ours: GPS + PoinTr** | **0.61** | **0.79** | **0.31** | **0.76** | **0.64** | **0.41** | **0.68** | **1.44** | **0.84** | **0.529** |
| SeedFormer | 0.72 | 0.81 | 0.40 | 0.89 | 0.71 | 0.50 | 0.77 | 1.49 | 0.92 | 0.472 |
| HyperCD + SeedFormer | 0.66 | 0.74 | 0.35 | 0.83 | 0.64 | 0.47 | 0.72 | 1.40 | 0.86 | 0.482 |
| InfoCD + SeedFormer | 0.65 | 0.72 | 0.31 | 0.81 | 0.62 | 0.43 | 0.71 | 1.38 | 0.84 | 0.490 |
| **Ours: GPS + SeedFormer** | **0.63** | **0.70** | **0.30** | **0.79** | **0.61** | **0.42** | **0.69** | **1.37** | **0.82** | **0.493** |

Table 12: Per-point L1-CD ×1000 (lower is better) on PCN.

| Methods | Plane | Cabinet | Car | Chair | Lamp | Couch | Table | Boat | Avg. |
|---|---|---|---|---|---|---|---|---|---|
| FoldingNet (Yang et al., 2018) | 9.49 | 15.80 | 12.61 | 15.55 | 16.41 | 15.97 | 13.65 | 14.99 | 14.31 |
| HyperCD + FoldingNet | 7.89 | 12.90 | 10.67 | 14.55 | 13.87 | 14.09 | 11.86 | 10.89 | 12.09 |
| InfoCD + FoldingNet | 7.90 | 12.68 | 10.83 | 14.04 | 14.05 | 14.56 | 11.61 | 11.45 | 12.14 |
| **Ours: GPS + FoldingNet** | **7.38** | **12.61** | **10.46** | **13.12** | **11.92** | **13.39** | **10.86** | **10.59** | **11.30** |
| PMP-Net (Wen et al., 2021) | 5.65 | 11.24 | 9.64 | 9.51 | 6.95 | 10.83 | 8.72 | 7.25 | 8.73 |
| HyperCD + PMP-Net | 5.06 | 10.67 | 9.30 | 9.11 | 6.83 | 11.01 | 8.18 | 7.03 | 8.40 |
| InfoCD + PMP-Net | 4.67 | 10.09 | 8.87 | 8.59 | 6.38 | 10.48 | 7.51 | 6.75 | 7.92 |
| **Ours: GPS + PMP-Net** | **4.52** | **10.02** | **8.80** | **8.45** | **6.31** | **10.42** | **7.46** | **6.70** | **7.84** |
| PoinTr (Yu et al., 2021b) | 4.75 | 10.47 | 8.68 | 9.39 | 7.75 | 10.93 | 7.78 | 7.29 | 8.38 |
| HyperCD + PoinTr | 4.42 | 9.77 | 8.22 | 8.22 | 6.62 | 9.62 | 6.97 | 6.67 | 7.56 |
| InfoCD + PoinTr | 4.06 | 9.42 | 8.11 | 7.81 | 6.21 | 9.38 | 6.57 | 6.40 | 7.24 |
| **Ours: GPS + PoinTr** | **4.03** | **9.39** | **8.03** | **7.78** | **6.18** | **9.33** | **6.56** | **6.38** | **7.21** |
| SnowflakeNet (Xiang et al., 2021) | 4.29 | 9.16 | 8.08 | 7.89 | 6.07 | 9.23 | 6.55 | 6.40 | 7.21 |
| HyperCD + SnowflakeNet | **3.95** | 9.01 | 7.88 | 7.37 | 5.75 | 8.94 | 6.19 | 6.17 | 6.91 |
| InfoCD + SnowflakeNet | 4.01 | 8.81 | 7.62 | 7.51 | 5.80 | 8.91 | 6.21 | 6.05 | 6.86 |
| **Ours: GPS + SnowflakeNet** | 3.97 | **8.79** | **7.61** | **7.47** | **5.73** | **8.85** | **6.15** | **6.03** | **6.82** |
| PointAttN (Wang et al., 2022) | 3.87 | 9.00 | 7.63 | 7.43 | 5.90 | 8.68 | 6.32 | 6.09 | 6.86 |
| HyperCD + PointAttN | 3.76 | 8.93 | 7.49 | 7.06 | 5.61 | 8.48 | 6.25 | 5.92 | 6.68 |
| InfoCD + PointAttN | 3.72 | 8.87 | 7.46 | 7.02 | 5.60 | 8.45 | 6.23 | 5.92 | 6.65 |
| **Ours: GPS + PointAttN** | **3.70** | **8.83** | **7.42** | **7.00** | **5.59** | **8.43** | **6.22** | **5.91** | **6.63** |
| SeedFormer (Zhou et al., 2022) | 3.85 | 9.05 | 8.06 | 7.06 | 5.21 | 8.85 | 6.05 | 5.85 | 6.74 |
| HyperCD + SeedFormer | 3.72 | 8.71 | 7.79 | 6.83 | 5.11 | 8.61 | 5.82 | 5.76 | 6.54 |
| InfoCD + SeedFormer | 3.69 | 8.72 | 7.68 | 6.84 | 5.08 | 8.61 | 5.83 | 5.75 | 6.52 |
| **Ours: GPS + SeedFormer** | **3.68** | **8.69** | **7.65** | **6.80** | **5.05** | **8.55** | **5.72** | **5.63** | **6.48** |

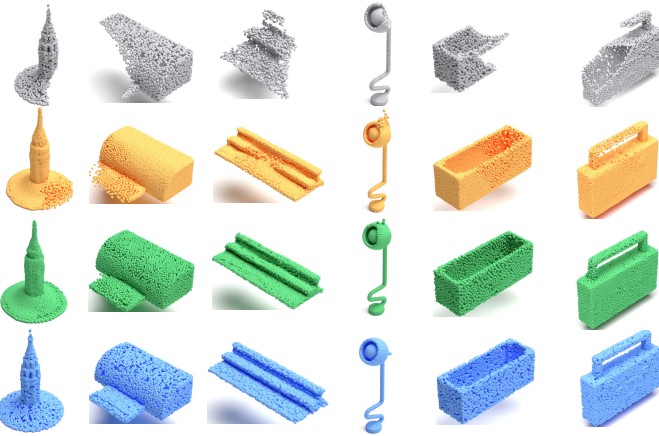

Figure 7: **Row-1:** Inputs of incomplete point clouds. **Row-2:** Outputs of Seedformer with CD. **Row-3:** Outputs of Seedformer with GPS. **Row-4:** Ground truth.

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
