# OpenReview forum: "GPS: A Probabilistic Distributional Similarity with Gumbel Priors for Set-to-Set Matching"
_ICLR.cc/2025/Conference — ICLR 2025 Poster_

### Official Review · Reviewer_YjK9 · 2024-11-01

**Soundness:** 3
**Presentation:** 2
**Contribution:** 3
**Rating:** 6
**Confidence:** 3

**Summary:**

This manuscript concentrates on the task of set-to-set matching. To this end, motivated by the observation that the distributions of minimum distances fro CD can be accurately modeled by Gumbel distribuctions, it proposes GPS based on Gumbel prior distributions. The proposed method has been validated on tasks including few-shot image classification and 3D point completion, and the results demonstrate its significance.

**Strengths:**

1. The idea of GPS is novel and interesting, expecially considering that metrics like Chamfer Distance have been widely adopted in many research areas and replacing Chamfer Distance with GPS maintains the same linear computational cost

2. The writing of this manuscript is acceptable

3. The proposed method has been validated on different tasks, and the experimental results demonstrate its significance

**Weaknesses:**

1. The organization of this paper can be further improved. For example, on the first page, Fig.2 is placed before Fig.1. Fig. 2 (c) is kind of confusing for me, even if I have read related context for several times.

2. In Section.3, Fig.3 doesn't help understand the methodology, but even make me more confusing.
There should have one image that helps understand the design of GPS better. In the current version, any of  Fig.1, Fig.2, Fig.3  doesn't help too much. I think there also lacks a figure demonstrating the whole paradigm of applying GPS to a specific task.

3. For the experimental part, I think the task of image matching (SuperGlue, CVPR 2020) or point cloud matching (Predator, CVPR 2021) should also be considered, as establishing correspondences is a fundamental problem in computer vision.

**Questions:**

See weaknesses.

---

> ### Author Response · Authors · 2024-11-25
>
> Thank you for the valuable comments. Below are our responses to your concerns:
>
> **1. Better Organization:** Figure 2(c) demonstrates that Gumbel distributions can effectively fit KNN data, not just first-order nearest neighbors. Figure 3 presents the graphical model for computing GPS, similar to those used in probabilistic modeling like pLSA and LDA. We will restructure the paper to enhance clarity.
>
> **2. Paradigm of Applying GPS to A Specific Task:** To utilize GPS, we can simply replace the set-to-set feature distances in the original approaches with our negative GPS scores and then train the networks. We will include some Python-like pseudo-code to illustrate this process.
>
> **3. More Experiments:** We conducted more experiments for both point cloud matching/registration and image matching using $GPS(\alpha, \beta)$ with only 1-NN and 1 Gumbel distribution. Details are as follows:
>
> * **(a) Point Cloud Matching with PREDATOR (Huang et al. CVPR, 2021):** We modified the public code from https://github.com/prs-eth/OverlapPredator by simply replacing the distances in the loss with the negative GPS scores. The results are listed below:
>
> | Samples          | 5000 | 2500 | 1000 | 500  | 250  |   Ave.$\pm$Std   |
>
> | PREDATOR    | 89.0  | 89.9  | 90.6  | 88.5 | 86.6 | 88.92$\pm$1.53 |
>
> | P + GPS(1, 1) | 89.3  | 89.6  | 90.0  | 88.2 | 88.7 | 89.16$\pm$0.72 |
>
> | P + GPS(1, 2) | 88.6  | 90.2  | 90.3  | 89.2 | 88.4 | 89.34$\pm$0.88 |
>
> | P + GPS(2, 2) | 89.4  | 89.4  | 90.8  | 89.3 | 89.3 | 89.64$\pm$0.65 |
>
> * **(b) Image Matching with LightGlue (Lindenberger et al. CVPR, 2023):** Replacing distances in the loss for SuperGlue (Sarlin et al. CVPR, 2020) with our negative GPS scores is not straightforward, because there are some numerical stability issues in the log-space. Instead, we utilized LightGlue (https://github.com/cvg/LightGlue), which is inherited from SuperGlue, for image matching. Same as we did for PREDATOR, we replaced the feature distances with our negative GPS scores for computing the loss in LightGlue. The results are listed below:
>
> | AUC | 5 degree | 10 degree | 20 degree | Ave.$\pm$Std |
>
> | LightGlue | 41.5 | 57.8 | 70.7 | 56.67$\pm$14.63 |
>
> | L + GPS(1, 1) | 42.5 | 48.3 | 71.2 | 57.33$\pm$14.37 |
>
> As we can see, in both applications, our GPS can significantly improve the baselines in terms of accuracy and robustness. Note that we have not fine-tuned our hyperparameters for GPS in the tables yet. In summary, our GPS can be applied to a wide range of applications for performance improvement.

---

> > ### Comment · Reviewer_YjK9 · 2024-11-27
> >
> > Thanks for your reply. I think the added experiments addressed my concerns well. I will keep my positive rating.

---

### Official Review · Reviewer_5tK4 · 2024-11-02

**Soundness:** 3
**Presentation:** 3
**Contribution:** 2
**Rating:** 6
**Confidence:** 4

**Summary:**

The paper proposes to calculate the similarity between sets based on the Gumbel prior distribution.

**Strengths:**

The presentation is nice and clear.
Very interesting set of experiments and they are extensive.

**Weaknesses:**

The GPS method proposed in the paper calculates the similarity between sets based on the Gumbel prior distribution, which is, to a certain extent, an improvement on existing methods. However, the degree of this innovation is relatively limited. For example, within the framework of set matching and similarity learning, many studies have attempted to introduce new methods and ideas from different perspectives. The GPS method, which is based on distance distribution fitting and the use of Gumbel distribution, has not fundamentally broken through the scope of existing research.

**Questions:**

There is a lack of in-depth theoretical discussion on some key issues. For example, in practical applications, the assumed independent and identically distributed conditions are difficult to strictly meet. The paper only shows that the assumption seems feasible through experimental results, but does not theoretically analyze the error range brought by this approximation and the potential impact on the performance of the method. This theoretical incompleteness may affect the reliability and universality of the method.

The experiments are mainly focused on a few common image classification and point cloud completion datasets. Although these datasets are representative, they cannot fully cover all possible application scenarios and data distributions.

Although the paper claims that the GPS method has achieved better results than some existing methods in the experiment, a careful observation of the experimental data shows that in most cases, the performance improvement is relatively small. For example, the accuracy improvement on some datasets may be only a few percentage points, which may not be significant in practical applications, especially considering the additional computational cost and model complexity that may be introduced.

**Details Of Ethics Concerns:**

Nothing

---

> ### Author Response · Authors · 2024-11-13
> **Clarification**
>
> We appreciate your comments and the insights. Before we offer a response, we were wondering whether you could clarify your comments.
>
> **Comparisons to Prior Works:** You noted that “within the framework of set matching and similarity learning, many studies have introduced new methods and ideas from different perspectives.” Could you point to specific studies or methods you feel would be most relevant for comparison.
>
> **Theoretical Discussion on Assumptions:** We noticed your point about the challenges of independent and identically distributed (i.i.d.) assumptions in practical applications. Could you clarify how you see this assumption impacting our methodology?
>
> **Performance Improvement Evaluation:** You comment that our method showed only modest accuracy improvements. We would appreciate if you could clarify what you mean with the context of other reviews here.

---

> > ### Comment · Reviewer_5tK4 · 2024-11-21
> > **Response**
> >
> > 1. In comparison, the following studies are related to this method:
> > Simmatch: Semi-supervised learning with similarity matching
> > Graph matching networks for learning the similarity of graph structured objects
> > Quasi-dense similarity learning for multiple object tracking
> > 2. This is fundamental to many traditional algorithms, but in real-world data, this assumption often does not hold because data is often not independent and identically distributed.
> > 3. The performance improvement is not significant, and I hope the author can give more interesting results.
> > Hope the author gives more insights to drive the consideration of the score. Thank you.

---

> > > ### Author Response · Authors · 2024-11-22
> > > **Any similarities to SimMatch or QDTrack is incidental. There is little overlap with GPS.**
> > >
> > > **GPS operates in a fundamentally different problem domain from both SimMatch and QDTrack**
> > >
> > > 1. GPS focuses on the general problem of set-to-set matching, a distinct area of research involving the comparison of distributions of points or embeddings between sets to evaluate their similarity. SimMatch focuses on semi-supervised learning through instance-level consistency; QDTrack, addresses object tracking by matching individual instances across video frames.
> > >
> > > 2. Set-to-set matching requires reasoning about global structure or distributional alignment between sets. For example, in point-cloud matching, GPS measures how well two distributions of 3D points align, an inherently set-level task. Such problems are outside the scope of instance-level methods like SimMatch or QDTrack, which cannot directly handle tasks involving statistical relationships across sets.
> > >
> > > 3. Set-to-Set matching has a rich history of research, with applications in diverse areas such as few-shot learning and 3D object reconstruction. GPS contributes to this domain by introducing a novel probabilistic framework enabling robust and efficient similarity modeling.
> > >
> > > **Example that clarifies this difference:**
> > > In point-cloud matching, a partially reconstructed 3D object (e.g., a car) is represented as a set of points in space. GPS measures the similarity between this point cloud and a complete reference model by comparing their distributions. This involves evaluating statistical alignment between sets of points, rather than classifying individual instances or tracking objects, as in SimMatch or QDTrack.

---

> ### Author Response · Authors · 2024-11-22
> **IID Assumption is not relevant to the paper - we mention it only to contextualize Gumbel.**
>
> We appreciate the reviewer’s comments regarding the i.i.d. assumption. The mention of i.i.d. in the paper is a side-comment meant to provide context for the use of the Gumbel distribution. It is not central to the methodology or results. The purpose of including it was to give a more complete explanation of the origins of the Gumbel distribution, but this was intended as an educational aside rather than a strict requirement of our approach.
>
> **Algorithmically**, our method simply involves:
>
> 1. Computing the nearest-neighbor distances between two sets.
> 2. Fitting a Gumbel distribution to these distances.
> 3. Using the resulting parameters to measure set-to-set similarity.
>
> The i.i.d. assumption does not constrain this process. The algorithm works effectively regardless of whether the data fully satisfies i.i.d. properties, as demonstrated by the empirical results (e.g., Figure 4). Indeed, the paper would be no different if we had omitted any reference to i.i.d., as the practical implementation is focused on fitting a Gumbel distribution to observed data.

---

> ### Author Response · Authors · 2024-11-22
> **Performance is SOTA**
>
> We appreciate the reviewer’s attention to the results and would like to clarify the contributions of our work and the evaluation of its performance. Specifically:
>
> **Relevance of Datasets and Baselines:** The datasets, baselines, and experimental tasks we consider are specifically chosen to align with the problem of set-to-set matching that GPS addresses. These benchmarks are widely recognized in the literature for evaluating methods in this domain, including:
> 1. Few-shot classification on datasets such as miniImageNet, tieredImageNet, and CIFAR-FS.
> 2. Point-cloud completion on ShapeNet-Part and KITTI.
>
> **State-of-the-Art Performance:**
> GPS achieves consistent state-of-the-art (SOTA) performance across benchmarks in few-shot classification and point-cloud completion tasks (see Table 5-12).
>
> **For example, in few-shot classification:**
> On the miniImageNet 1-shot 5-way classification task, GPS achieves an accuracy of 66.27 ± 0.37%, compared to 64.01 ± 0.32% by InfoCD (NeuRIPS 2023), 63.63 ± 0.65% by HyperCD (CVPR 2023), and 63.36 ± 0.75% by DeepEMD. This represents an improvement of up to 2.26 percentage points over prior SOTA methods. On tieredImageNet 1-shot 5-way classification, GPS achieves 73.16 ± 0.43%, outperforming HyperCD (70.58 ± 0.81%) and InfoCD (70.97 ± 0.59%) by similar margins .
>
> **CIoud completion**: On the ShapeNet-Part dataset, GPS achieves the lowest L2-CD score (3.94 ± 0.003) compared to 4.01 ± 0.004 by InfoCD and 4.03 ± 0.007 by HyperCD . On the KITTI dataset, GPS achieves a Fidelity of 1.883 and an MMD of 0.302, consistently outperforming prior methods.
>
> **Significance of Improvement - Improvements of 1-2% in these tasks are considered highly significant due to the competitive nature of existing baselines. For instance:
>
> **Novel Contributions:** Beyond performance improvements, GPS introduces a novel probabilistic framework for set-to-set similarity, leveraging the Gumbel distribution to capture distributional alignment efficiently. This innovation addresses computational challenges in previous approaches while maintaining high accuracy.
>
> **Request for Clarification:** We respectfully request the reviewer to clarify what is meant by "not significant" and to provide specific criteria or examples of what they would consider "more interesting results." Our evaluation rigorously benchmarks against all prior works, and GPS demonstrates clear SOTA performance.

---

> ### Comment · Reviewer_5tK4 · 2024-11-23
> **Thank you**
>
> Thanks to the author for his patient reply, which has solved most of my problems. I will improve my score. I am curious about how this method works in the field of point cloud registration. I noticed that point cloud registration is a research field with many applications. Here are some references.
>
> RIGA: Rotation-Invariant and Globally-Aware Descriptors for Point Cloud Registration, TPAMI 2024
>
> Rotation-Invariant Transformer for Point Cloud Matching, CVPR 2023
>
> EGST: Enhanced geometric structure transformer for point cloud registration TVCG 2024
>
> TEASER: Fast and Certifiable Point Cloud Registration, TRO 2021
>
> 3D Registration With Maximal Cliques, CVPR 2023

---

> > ### Comment · Reviewer_5tK4 · 2024-11-25
> > **Point cloud registration**
> >
> > Could you please give me some results on point cloud registration?

---

> > ### Author Response · Authors · 2024-11-25
> > **Thank you!**
> >
> > Thank you for increasing your score!
> >
> > We have conducted additional experiments on both image matching and point cloud matching/registration. The results consistently show significant improvements in performance and robustness compared to baseline approaches. We hope these findings demonstrate that our GPS loss function for deep learning is novel and has the potential to impact many applications involving set-to-set matching.
> >
> > So, would you like to further increase your score :)

---

> > > ### Comment · Reviewer_5tK4 · 2024-11-30
> > >
> > > Thank you for the additional experiments and the clarifications provided. The improvements in performance and robustness in both image and point cloud matching are indeed promising. I acknowledge the potential impact of your GPS loss function in set-to-set matching applications. However, further discussion on the specific challenges addressed by your method would be helpful to fully assess its novelty and broader applicability. I need more reasons to raise my score again.

---

> > > > ### Author Response · Authors · 2024-11-30
> > > > **Further clarification needed for your comment!**
> > > >
> > > > We thank the reviewer for the feedback!
> > > >
> > > > **Clarification Is Needed:** What does "further discussion on the specific challenges addressed by your method" mean? What specific questions do you have? We would like to address your questions if you can describe them clearly.
> > > >
> > > > **Core Contribution:**
> > > > Our work addresses the problem of set-to-set matching in a deep learning context. Specifically, we compute for instance 1-nearest-neighbor distances between two sets and parameterize these distances using the Gumbel distribution. This parameterization enables backpropagation through extrema, allowing us to train neural networks effectively for set-to-set similarity tasks.
> > > >
> > > > *To the best of our knowledge, this is the first work to introduce the Gumbel similarity as a differentiable loss function for set-to-set matching in neural networks. Our primary goal is to provide a practical solution to this problem, as evidenced by the empirical results.*

---

> > > > > ### Author Response · Authors · 2024-12-02
> > > > > **More results**
> > > > >
> > > > > Dear Reviewer,
> > > > >
> > > > > We have run more experiments using different distributions on **point cloud completion** using the setting in Table 8, and list the results below:
> > > > >
> > > > > | Distribution | L2-CD$\times 10^3$ |
> > > > >
> > > > > | Weibull | 4.09 |
> > > > >
> > > > > | Normal | 4.17 |
> > > > >
> > > > > | Logistic | 4.18 |
> > > > >
> > > > > | Log-Logistic | 4.10 |
> > > > >
> > > > > | Chi-Squared | 4.68 |
> > > > >
> > > > > | Gamma | 4.22 |
> > > > >
> > > > > **| GPS (Gumbel) | 3.94 |**
> > > > >
> > > > > We further run these distributions on **few-shot image classification** using the setting in Table 1, and list the results below:
> > > > >
> > > > > | Distribution | Accuracy (%) |
> > > > >
> > > > > | Weibull | 72.59 |
> > > > >
> > > > > | Normal | 72.85 |
> > > > >
> > > > > | Logistic | 72.59 |
> > > > >
> > > > > | Log-Logistic | 72.64 |
> > > > >
> > > > > | Chi-Squared | 72.89 |
> > > > >
> > > > > | Gamma | 72.65 |
> > > > >
> > > > > **| GPS (Gumbel) | 74.22 |**
> > > > >
> > > > > As you can see, our GPS consistently outperforms other methods across these tasks. We believe that these additional results, along with the evidence in Figure 2 and our previous discussion, can further demonstrate the novelty and broader applicability of our GPS approach.
> > > > >
> > > > > **Is there anything else you would like us to address, as today is the rebuttal deadline? Thank you.**

---

> > > > > > ### Comment · Reviewer_5tK4 · 2024-12-03
> > > > > >
> > > > > > The author has addressed all of my concerns. I have no further questions and I will be raising my score. Good luck.

---

> ### Author Response · Authors · 2024-11-25
> **Thanks for your patience for more results**
>
> As suggested by Reviewer auE8, Reviewer S2Z9 and Reviewer YjK9, we conducted more experiments for both point cloud matching and image matching using $GPS(\alpha, \beta)$ with only 1-NN and 1 Gumbel distribution. Details are as follows:
>
> 1. **Point Cloud Matching with PREDATOR (Huang et al. CVPR, 2021):** We modified the public code from https://github.com/prs-eth/OverlapPredator by simply replacing the distances in the loss with the negative GPS scores. The results are listed below:
>
> | Samples          | 5000 | 2500 | 1000 | 500  | 250  |   Ave.$\pm$Std   |
>
> | PREDATOR    | 89.0  | 89.9  | 90.6  | 88.5 | 86.6 | 88.92$\pm$1.53 |
>
> | P + GPS(1, 1) | 89.3  | 89.6  | 90.0  | 88.2 | 88.7 | 89.16$\pm$0.72 |
>
> | P + GPS(1, 2) | 88.6  | 90.2  | 90.3  | 89.2 | 88.4 | 89.34$\pm$0.88 |
>
> | P + GPS(2, 2) | 89.4  | 89.4  | 90.8  | 89.3 | 89.3 | 89.64$\pm$0.65 |
>
> 2. **Image Matching with LightGlue (Lindenberger et al. CVPR, 2023):** Replacing distances in the loss for SuperGlue (Sarlin et al. CVPR, 2020) with our negative GPS scores is not straightforward, because there are some numerical stability issues in the log-space. Instead, we utilized LightGlue (https://github.com/cvg/LightGlue), which is inherited from SuperGlue, for image matching. Same as we did for PREDATOR, we replaced the feature distances with our negative GPS scores for computing the loss in LightGlue. The results are listed below:
>
> | AUC | 5 degree | 10 degree | 20 degree | Ave.$\pm$Std |
>
> | LightGlue | 41.5 | 57.8 | 70.7 | 56.67$\pm$14.63 |
>
> | L + GPS(1, 1) | 42.5 | 48.3 | 71.2 | 57.33$\pm$14.37 |
>
> As we can see, in both applications, our GPS can significantly improve the baselines in terms of accuracy and robustness. Note that we have not fine-tuned our hyperparameters for GPS in the tables yet. In summary, our GPS can be applied to a wide range of applications for performance improvement.

---

### Official Review · Reviewer_S2Z9 · 2024-11-02

**Soundness:** 3
**Presentation:** 3
**Contribution:** 3
**Rating:** 8
**Confidence:** 2

**Summary:**

It proposes a novel probabilistic method for set-to-set matching called GPS (Gumbel Prior Similarity), based on distributional similarity and Gumbel distributions. This method measures the similarity between the underlying distributions generating the sets. This research is very meaningful and contributes a lot.

**Strengths:**

1. The method presents an innovative probabilistic approach to set-to-set matching based on distributional similarity and Gumbel distributions. The idea is quite novel and significant, as it is helpful for the training of differentiable neural networks and offers considerable value to the community.

2. The experiments look promising, especially regarding the superior performance in few-shot image classification and 3D point cloud completion.

3. The paper is well-written and easy to follow.

**Weaknesses:**

I don't see any obvious drawbacks, but I am concerned about the efficiency issues mentioned in your limitations.

**Questions:**

I have some suggestions. The proposed method can perform better in set-to-set matching, demonstrating good performance in few-shot image classification and 3D point cloud completion. I am particularly interested in whether it can also achieve good performance in image matching and point cloud matching, as they also involve the set-to-set matching problem. If it performs well in these areas, it would further highlight the impact of your method. If possible, please consider adding relevant experiments.

---

> ### Author Response · Authors · 2024-11-25
>
> We thank the reviewer for the valuable comments. We have conducted more experiments for both point cloud matching/registration and image matching using $GPS(\alpha, \beta)$ with only 1-NN and 1 Gumbel distribution. Details are as follows:
>
> **(1) Point Cloud Matching with PREDATOR (Huang et al. CVPR, 2021):** We modified the public code from https://github.com/prs-eth/OverlapPredator by simply replacing the distances in the loss with the negative GPS scores. The results are listed below:
>
> | Samples          | 5000 | 2500 | 1000 | 500  | 250  |   Ave.$\pm$Std   |
>
> | PREDATOR    | 89.0  | 89.9  | 90.6  | 88.5 | 86.6 | 88.92$\pm$1.53 |
>
> | P + GPS(1, 1) | 89.3  | 89.6  | 90.0  | 88.2 | 88.7 | 89.16$\pm$0.72 |
>
> | P + GPS(1, 2) | 88.6  | 90.2  | 90.3  | 89.2 | 88.4 | 89.34$\pm$0.88 |
>
> | P + GPS(2, 2) | 89.4  | 89.4  | 90.8  | 89.3 | 89.3 | 89.64$\pm$0.65 |
>
> **(2) Image Matching with LightGlue (Lindenberger et al. CVPR, 2023):** Replacing distances in the loss for SuperGlue (Sarlin et al. CVPR, 2020) with our negative GPS scores is not straightforward, because there are some numerical stability issues in the log-space. Instead, we utilized LightGlue (https://github.com/cvg/LightGlue), which is inherited from SuperGlue, for image matching. Same as we did for PREDATOR, we replaced the feature distances with our negative GPS scores for computing the loss in LightGlue. The results are listed below:
>
> | AUC | 5 degree | 10 degree | 20 degree | Ave.$\pm$Std |
>
> | LightGlue | 41.5 | 57.8 | 70.7 | 56.67$\pm$14.63 |
>
> | L + GPS(1, 1) | 42.5 | 48.3 | 71.2 | 57.33$\pm$14.37 |
>
> As we can see, in both applications, our GPS can significantly improve the baselines in terms of accuracy and robustness. Note that we have not fine-tuned our hyperparameters for GPS in the tables yet. In summary, our GPS can be applied to a wide range of applications for performance improvement.

---

> > ### Comment · Reviewer_S2Z9 · 2024-12-03
> > **Thanks for your response.**
> >
> > I think it's a good work. The author has also addressed all of my concerns. Therefore, I keep the high score unchanged.

---

### Official Review · Reviewer_45Az · 2024-11-04

**Soundness:** 3
**Presentation:** 3
**Contribution:** 2
**Rating:** 6
**Confidence:** 4

**Summary:**

The paper presents GPS, a Gumbel-prior-distribution mechanism to train models that can compare sets and compute correspondences. The idea is to use the Gumbel distribution to model extrema (minima or maxima) of the matching scores; this is based on Extreme Value Theory. GPS also proposes the use of an offset to compensate the modeled scores and improve the modeling of the scores. The paper presents experiments on image classification and point cloud completion where the proposed method consistently show improvements over the considered baselines.

**Strengths:**

- Motivation and description of the problem is quite clear and it is quite important. I agree that we need to investigate better solution for set-to-set matching. In particular, I agree that the use of the statistical Extreme Value Theory into ML/AI can bring benefits to many problems.
- The clarity of the paper in general is clear. The narrative is clear and I think the paper can be reproduce w/ some reasonable effort.
- The set of experiments is adequate and tackles different applications showing the benefits and applicability of the approach.

**Weaknesses:**

While I agree about the use of EVT or Gumbel distribution to tackle problems that require modeling extrema (minimum or maximum) of matching scores, I disagree with the use of particularly using the Gumbel distribution only. Here are the specific concerns:

1. According to EVT, the distribution that generally models the extrema (minima or maxima) is the Generalized Extreme Value (GEV) distribution; the Gumbel distribution is a special case of the GEV distribution. Thus, it is possible that for some problems the Gumbel distribution is not the appropriate one and thus it can affect performance. Note however that EVT in general can only be applied when the number of scores is large, otherwise, the theory and thus the distributions of EVT (including Gumbel) cannot be applied, strictly speaking. This is something that I think the paper is not exploring nor stating.

2. While it is true that Gumbel can model minima or maxima, it is not true that it can model order statistics. In other words, I think the narrative of the paper is correct only when it states that it models minima or maxima when comparing entities of a set. However, the Gumbel distribution cannot model order statistics, i.e., statistics of the 2nd, 3rd, or Kth scores. Given this, I find concerning that the paper uses it to model the k-th nearest neighbors using a mixture of Gumbel distributions (lines 200 - 203). EVT only models the minima or maxima, but not the order statistics. I don't see a justification in the paper describing rigorously the use of Gumbel distribution in this case. Unless, the introduction of $\delta$ as shown in line 232 compensates for that. But unfortunately the paper does not justifies this well.

3. The paper is missing prior work exploring EVT to model minima/maxima in matching procedures. Thus, I think the narrative overstates that this is the first work using statistical information for set-to-set matching. See references shown below.

4. Incomplete experiments. While I think the applications used in the experiments are diverse, I think the experiments would've been more informative and more convincing if they could show that other models (e.g., a transformer, or CLIP) also benefit from the proposed GPS.

References:
1. Scheirer, Walter J., et al. "Meta-recognition: The theory and practice of recognition score analysis." IEEE transactions on pattern analysis and machine intelligence 33.8 (2011): 1689-1695.
2. Fragoso, Victor, et al. "EVSAC: accelerating hypotheses generation by modeling matching scores with extreme value theory." Proceedings of the IEEE international conference on computer vision. 2013.
3. Fragoso, Victor, and Matthew Turk. "SWIGS: A swift guided sampling method." Proceedings of the IEEE Conference on Computer Vision and Pattern Recognition. 2013.

-------
Post-discussion

After discussion with the authors, clarity about the motivation of using Gumbel distribution improved. I still think the paper could've  included a more solid justification, but the experiments show the method works. I would just recommend the authors to update the narrative justifying the design choices a bit better and I will raise my score.

**Questions:**

1. Why is the use of $\delta$ necessary from the theoretical point of view?

---

> ### Author Response · Authors · 2024-11-25
>
> We thank the reviewer for the valuable comments. Below are our responses to your concerns:
>
> **1. Generalized Extreme Value (GEV) Distribution vs. Gumbel Distribution:** As mentioned, GEV is a generalization of Gumbel distributions by incorporating additional hyperparameters. In our research, we have tested various GEV distributions with different hyperparameters (though these experiments were not included in the submission). We found that they did not significantly improve performance compared to the Gumbel distribution but only increased the fine-tuning time. Therefore, to maintain simplicity, we chose the Gumbel distribution as our prior distribution for fitting data. We will add such discussion in our final paper.
>
> **2. Modelling ordinary order statistics using Gumbel:** Typically, the Gumbel distribution is used to model minima or maxima, such as first-order nearest neighbors in our case. However, we would like to emphasize the following points:
>
> * (a) Theoretically, it has been shown that under certain conditions, Extreme Value Theory (EVT) can be applied to (generalized) order statistics (e.g., Nasri-Roudsari, Dirk. "Extreme value theory of generalized order statistics." Journal of Statistical Planning and Inference 55.3 (1996): 281-297).
>
> * (b) Empirically, we have demonstrated that it is possible to fit KNN data with Gumbel distributions, as shown in Fig. 2(c).
>
> * (c) Our approach is similar to Gaussian Mixture Models (GMM), where GMM uses Gaussian as a prior, while our GPS uses Gumbel as a prior. The actual distribution of the data (whether Gaussian or Gumbel) is less important as long as the problem can be effectively addressed based on the model.
>
> **3. Related References:** Thank you for providing these references; we will include them in our paper. However, the main point we want to emphasize is that we are **the first to introduce the Gumbel similarity as a set-to-set loss function in deep learning**, rather than using it for matching. To the best of our knowledge, no existing work has done this, and we would appreciate any feedback you may have.
>
> **4. Experiments:** A closer reading of our submission, specifically lines 152-155, reveals that **SnowflakeNet, PoinTr, and SeedFormer are all transformer-based networks** for point cloud completion, and our GPS consistently enhances these baseline approaches.
>
> Moreover, we conducted more experiments for both point cloud matching/registration and image matching using $GPS(\alpha, \beta)$ with only 1-NN and 1 Gumbel distribution. Details are as follows:
>
> * **(a) Point Cloud Matching with PREDATOR (Huang et al. CVPR, 2021):** We modified the public code from https://github.com/prs-eth/OverlapPredator by simply replacing the distances in the loss with the negative GPS scores. The results are listed below:
>
> | Samples          | 5000 | 2500 | 1000 | 500  | 250  |   Ave.$\pm$Std   |
>
> | PREDATOR    | 89.0  | 89.9  | 90.6  | 88.5 | 86.6 | 88.92$\pm$1.53 |
>
> | P + GPS(1, 1) | 89.3  | 89.6  | 90.0  | 88.2 | 88.7 | 89.16$\pm$0.72 |
>
> | P + GPS(1, 2) | 88.6  | 90.2  | 90.3  | 89.2 | 88.4 | 89.34$\pm$0.88 |
>
> | P + GPS(2, 2) | 89.4  | 89.4  | 90.8  | 89.3 | 89.3 | 89.64$\pm$0.65 |
>
> * **(b) Image Matching with LightGlue (Lindenberger et al. CVPR, 2023):** Replacing distances in the loss for SuperGlue (Sarlin et al. CVPR, 2020) with our negative GPS scores is not straightforward, because there are some numerical stability issues in the log-space. Instead, we utilized LightGlue (https://github.com/cvg/LightGlue), which is inherited from SuperGlue, for image matching. Same as we did for PREDATOR, we replaced the feature distances with our negative GPS scores for computing the loss in LightGlue. The results are listed below:
>
> | AUC | 5 degree | 10 degree | 20 degree | Ave.$\pm$Std |
>
> | LightGlue | 41.5 | 57.8 | 70.7 | 56.67$\pm$14.63 |
>
> | L + GPS(1, 1) | 42.5 | 48.3 | 71.2 | 57.33$\pm$14.37 |
>
> As we can see, in both applications, our GPS can significantly improve the baselines in terms of accuracy and robustness. Note that we have not fine-tuned our hyperparameters for GPS in the tables yet. In summary, our GPS can be applied to a wide range of applications for performance improvement.
>
> **5. Usage of $\delta$ from a Theoretical View:** Theoretically, the constant $\delta$ influences the mode of a Gumbel distribution, as shown in Fig. 4. As mentioned in lines 258-260, "In many applications, such as point cloud completion, the objective is to minimize the distances between predictions and the ground truth." In these applications, this constant distance shift is crucial for learning.

---

> > ### Comment · Reviewer_45Az · 2024-11-25
> > **RE: Official Comment by Authors**
> >
> > **1.** I suspect GEV (or any other special-case distribution) is not providing good results since you need a long iid sequence from the same process. Nevertheless, I think it would've been great to add those experiments to the supplemental or in this rebuttal as evidence to make the argument much stronger. Given the lack of evidence of these experiments w/ different distributions, I get more questions than answers. What distributions did you try? How long were the sequences? What parameter estimator did you use?
> >
> > **2(a)** It would've been useful to know the conditions, and if those conditions apply to this work.
> >
> > **2(b)** I agree that empirically the results support that Gumbel provides good results. However, it is unclear at the moment if other distributions can perform better. Given that the theoretical argument of the paper is not that solid, it would've been great to show experiments showing that Gumbel indeed was the best special case.
> >
> > **2(c)** If the actual distribution of the data is not important, then I would recommend changing the theoretical narrative of the paper. It is not agreeing with the experiments and thus causing more confusion.
> >
> > **3.** Please then explicitly mention this in the paper, and perhaps consider changing the title of the paper.
> > **4.(a)** It is good to see the approach perform comparably or statistically equivalent than PREDATOR.
> > **4.(b)** It is also good to see that the proposed approach improved, however, the improvement is not significant as claimed in the answer.
> >
> > **5.** Thanks for clarifying. Please include this explanation in a final manuscript to improve clarity.

---

> > > ### Author Response · Authors · 2024-11-26
> > > **Clarification: Paper's focus is practical - i.e. Gumbel-Based Set Matching, Not EVT Extensions**
> > >
> > > We appreciate the reviewer’s thoughtful comments regarding the relevance of EVT and the Gumbel distribution. In this post we would like to clarify our focus because we feel it is getting lost in the EVT discussion. While EVT provided inspiration for our approach (and we will cite these references in the final version), we would like to clarify the focus of our contribution:
> > >
> > > **Core Contribution:**
> > > Our work addresses the problem of set-to-set matching in a deep learning context. Specifically, we compute for instance 1-nearest-neighbor distances between two sets and parameterize these distances using the Gumbel distribution. This parameterization enables backpropagation through extrema, allowing us to train neural networks effectively for set-to-set similarity tasks.
> > >
> > > *To the best of our knowledge, this is the first work to introduce the Gumbel similarity as a differentiable loss function for set-to-set matching in neural networks. Our primary goal is to provide a practical solution to this problem, as evidenced by the empirical results.*
> > >
> > > **On EVT’s Role:**
> > > While EVT provides a theoretical foundation for modeling extrema, our focus is not on advancing EVT theory. Instead, we use the Gumbel distribution as a practical and effective tool for modeling 1-NN distances. Our view is that whether EVT could rigorously justify our method or whether alternative distributions (e.g., GEV) might perform better is an interesting direction for future work.
> > >
> > > **Why Gumbel?**
> > > The Gumbel distribution was chosen because:
> > >
> > > 1. It effectively models extrema in our context (e.g., 1-NN distances).
> > > 2. It enables a differentiable parameterization, which is essential for training neural networks.
> > > 3. It provides robust empirical performance, as demonstrated in our results.
> > >
> > > While alternative distributions (e.g., Weibull, Fréchet) are possible, we found that they introduced additional complexity (eg. shape parameter).
> > >
> > > **Future Directions:**
> > > We agree with the reviewer that further exploration of EVT and alternative models is an exciting avenue for future research. However, the focus of this work is on solving the practical problem of set-to-set matching in a deep learning context, which our method addresses effectively.

---

> > > > ### Comment · Reviewer_45Az · 2024-11-26
> > > > **RE: Clarification**
> > > >
> > > > I am not focusing on the EVT discussion, all I am focusing on is to get a solid justification of design choices of your proposed method. Clearly, EVT was one justification but unfortunately experiments are not supporting it. What I am really saying is if EVT is not the right justification, then at least showing an empirical study comparing different distributions to model the the 1-NN distances should be included to make the narrative solid.

---

> > > > > ### Author Response · Authors · 2024-11-26
> > > > > **Rebuttal: Clarifying Design Choices in GPS**
> > > > >
> > > > > We thank the reviewer for raising an important point regarding the justification of design choices in GPS, particularly the use of Gumbel distributions and the focus on distributional similarity. Below, we provide clarification and seek further input to ensure our narrative aligns with the reviewer’s expectations.
> > > > >
> > > > > **On EVT and Empirical Justification** As the reviewer notes, EVT provided an initial theoretical motivation for using Gumbel distributions to model extrema such as 1-NN distances. However, we agree that relying solely on EVT is insufficient without stronger empirical evidence. Our use of the Gumbel distribution is driven by the following considerations:
> > > > >
> > > > > a. **Alignment with Extrema:** Gumbel distributions are specifically designed to model minima or maxima, making them a natural fit for 1-NN, 2-NN, and 3-NN distances, which represent extrema in the set-to-set matching problem.
> > > > >
> > > > > b. **Practical Effectiveness:** Gumbel’s differentiable parameterization allows backpropagation through extrema, which is critical for training neural networks.
> > > > >
> > > > > *To further address the reviewer’s request for empirical validation, we plan to include additional experiments in the supplementary material. These will compare Gumbel to alternative distributions (e.g., Gaussian, Weibull) in terms of:*
> > > > >
> > > > > i. Fit quality: How well each distribution models 1-NN distances;
> > > > >
> > > > > ii. Task performance: The impact of these distributions on the downstream tasks of few-shot classification and point-cloud completion.
> > > > >
> > > > > We hope this additional evidence will clarify why Gumbel was chosen.
> > > > >
> > > > > **On Distributional Similarity:**
> > > > > The reviewer may be referring to the broader concept of distributional similarity, which is central to GPS. Traditional metrics like Chamfer Distance (CD) and Earth Mover’s Distance (EMD) are inherently point-based and do not explicitly capture the statistical relationships between sets. GPS addresses this limitation by:
> > > > >
> > > > > i. Modeling the distributions of nearest-neighbor distances (1-NN, 2-NN, etc.) probabilistically using Gumbel.
> > > > >
> > > > > ii. Representing these distances in a way that reflects the underlying distributional similarity between sets, rather than isolated point matches.
> > > > >
> > > > > We would appreciate clarification from the reviewer if this aspect of distributional modeling aligns with their concern or if additional comparisons to traditional metrics like InfoCD are necessary to strengthen this point.
> > > > >
> > > > > **Seeking Reviewer Feedback:**
> > > > > While we aim to address the reviewer’s concerns through supplementary experiments and narrative improvements, we would greatly benefit from additional clarification:
> > > > >
> > > > > Is the reviewer primarily concerned with the choice of Gumbel versus other distributions, or the broader framework of distributional similarity introduced by GPS? Would the proposed comparative study (e.g., Gumbel vs. Gaussian or Weibull) sufficiently address the concern about design choices? Are there specific distributions or metrics the reviewer believes would be more relevant for comparison?

---

> ### Author Response · Authors · 2024-11-26
> **Some results using Fréchet and Weibull distributions**
>
> Consider the PDF of a Generalized Extreme Value distribution, $GEV(\mu, \sigma, \xi)$, as follows:
>
> $$
> f(x) = \frac{1}{\sigma} t(y)^{\xi+1} \exp(-t(y)),
> $$
> where
> $$y = \frac{x - \mu}{\sigma}$$, $$t(y) = \exp(-y)$$ if $\xi=0$, otherwise, $$t(y) = (1+\xi y )^{-\frac{1}{\xi}}$$. Here, $x\in\mathbb{R}$ denotes a random variable, and $\mu\in\mathbb{R}, \sigma>0, \xi\in\mathbb{R}$ are three parameters for the GEV. Note that the shape parameter $\xi$ controls the distribution family, i.e., $\xi=0$ leads to a Gumbel distribution, $\xi>0$ leads to a Fréchet distribution, and $\xi<0$ leads to a Weibull distribution.
>
> We reran the experiments in Table 4 for few-shot image classification on CIFAR-FS dataset, with $\mu=0.8, \sigma=0.5$ (equivalently $\alpha=5, \beta=2$ in our submission). The table below summarizes the comparison results between distributions:
>
> | Weibull | Gumbel | Fréchet |
>
> | $\xi=-1$ | $\xi=-0.1$ | $\xi=-0.01$ | $\xi=0$ | $\xi=0.01$ | $\xi=0.1$ | $\xi=1$ |
>
> | 71.29 | 73.18 | 73.47 | 74.22 | 73.65 | 73.31 |71.78 |
>
> The trend of such numbers is clear that the accuracy seems higher towards $\xi=0$, i.e. Gumbel. We hope that such results can convince the reviewer that Gumbel distributions may be a better choice to fit the nearest neighbors than the other two distributions for learning, not only higher accuracy but also fewer parameters.

---

> > ### Comment · Reviewer_45Az · 2024-11-26
> > **RE: Some results using Fréchet and Weibull distributions**
> >
> > Thanks for providing these results. However, given that EVT clearly is not working as expected, why not considering other distributions? E.g., Gaussian? Have you visualized the histograms of the distances? Then decide what distribution to use?
> > This is my concern. It seems like EVT is still being considered but clearly your experiments are telling you EVT distributions may not be a great family to fit your data. Thus, my question is what other distributions beyond GEV family can we use?

---

> ### Author Response · Authors · 2024-11-26
> **Rebuttal: Justifying the Design Choice of Gumbel Distribution**
>
> We thank the reviewer for their thoughtful feedback and for encouraging deeper consideration of alternative distributions. First of all, we need to highlight that Figure 2 indeed plots the distributions, which the reviewer may have missed looking at. Below, we clarify why the Gumbel distribution remains the most suitable choice for GPS and address the reviewer’s suggestions:
>
> **1.	Empirical Evidence from Figure 2:** Figure 2 shows the histograms of 1-NN, 2-NN, and 3-NN distances exhibit:
>
> * Skewed, heavy-tailed behavior, which aligns naturally with the Gumbel distribution.
>
> * Zero derivatives near zero, reflecting a smooth plateau that is not captured by distributions like Gamma or Erlang, which have positive slopes near zero.
>
> **2.	Why Not Gaussian, Exponential, or Gamma:**
>
> * Gaussian: Assumes symmetry and light tails, which contradicts the observed data’s skewness and heavy tails.
>
> * Exponential: Lacks flexibility to model broader variability or smooth behavior near zero.
>
> * Gamma/Erlang: While flexible for skewed data, these distributions are not designed for modeling extrema and cannot reproduce the zero-derivative property without significant parameter tuning.
>
> **3.	Why Gumbel:**
>
> * The Gumbel distribution is explicitly designed for modeling extrema and aligns with the observed characteristics of the data, including skewness, heavy tails, and smoothness near zero.
>
> * Its differentiable parameterization allows backpropagation through extrema, enabling efficient gradient-based optimization in neural network training.
>
> **4.	Additional Results:** We have run some distributions on **point cloud completion** using the setting in Table 8, and list the results below:
>
> | Distribution | L2-CD$\times 10^3$ |
>
> | Weibull | 4.09 |
>
> | Normal | 4.17 |
>
> | Logistic | 4.18 |
>
> | Log-Logistic | 4.10 |
>
> | Chi-Squared | 4.68 |
>
> | Gamma | 4.22 |
>
> **| GPS (Gumbel) | 3.94 |**
>
> We further run some distributions on **few-shot image classification** using the setting in Table 1, and list the results below:
>
> | Distribution | Accuracy (%) |
>
> | Weibull | 72.59 |
>
> | Normal | 72.85 |
>
> | Logistic | 72.59 |
>
> | Log-Logistic | 72.64 |
>
> | Chi-Squared | 72.89 |
>
> | Gamma | 72.65 |
>
> **| GPS (Gumbel) | 74.22 |**
>
> We believe that this additional analysis, combined with the evidence in Figure 2, should be able to further clarify why Gumbel remains the most effective choice for GPS. We appreciate the reviewer’s feedback and look forward to further input.

---

> ### Author Response · Authors · 2024-12-02
>
> Dear Reviewer,
>
> To address your concerns, we have included additional results using various distributions for both few-shot image classification and point cloud completion. Our GPS consistently performs best across these tasks. These results, along with our discussion on different distribution choices, have clearly demonstrated our novelty.
>
> **Is there anything else you would like us to address, as today is the rebuttal deadline? Thank you.**

---

### Official Review · Reviewer_auE8 · 2024-11-06

**Soundness:** 3
**Presentation:** 3
**Contribution:** 3
**Rating:** 8
**Confidence:** 2

**Summary:**

The paper presents a new similarity metric for set-to-set matching problems. The proposed similarity metric leverages Gumbel distribution as a prior to model k-NN across sets. This formulation makes the proposed similarity metric more robust to outliers compared to other similarity metrics like Chamfer Distance while being significantly more efficient to compute compared to robust metrics like Chamfer Distance.

The proposed similarity metric is applied to model objective functions for two problem settings: few-shot classification, point-cloud completion. The proposed metric outperforms other metrics across all evaluations and is similar to Chamfer Distance in runtime.

**Strengths:**

- Set-to-set matching is fundamental to many problem settings, the proposed formulation is novel, theoretically well-motivated and empirically outperforms other similarity metrics in chosen application domains. This is a significant contribution of interest to the wider community and can have impact beyond the two applications.
- Extensive evaluation and transparent reporting of influence of hyperparamters on results.
- I highly appreciate code submission.

**Weaknesses:**

- As authors have already pointed out, searching for hyperparameters for the proposed formulation is non-trivial. This might limit wider adoption of this metric.

**Questions:**

- Have you considered applying the metric for matching and registration problems such as point cloud registration? The paper mentions that "Set-to-set matching aims to identify correspondences between two sets of unordered items by minimizing a distance metric ... " however it is not clear to me how the proposed metric can be used to yield such correspondences. It might be valuable to add some discussion on potential formulations for such applications.

---

> ### Author Response · Authors · 2024-11-25
>
> Thank you for your valuable comments! Below are our responses to your questions:
>
> **1. Hyperparameter Tuning:** In fact, we found that this issue is moderate in practice for different applications, because (1) we can always choose the nearest neighbors for GPS and (2) the ranges for $\alpha, \beta$ are often very limited.
>
> **2. Finding correspondences:** This process is automatically handled by the nearest neighbor search when computing GPS, requiring no additional steps. As training with GPS progresses, we anticipate that the correspondences between the sets will improve over time, resulting in enhanced performance.
>
> **3. Point cloud registration:** We conducted more experiments for both point cloud matching/registration and image matching using $GPS(\alpha, \beta)$ with only 1-NN and 1 Gumbel distribution. Details are as follows:
>
> * **(a) Point Cloud Matching with PREDATOR (Huang et al. CVPR, 2021):** We modified the public code from https://github.com/prs-eth/OverlapPredator by simply replacing the distances in the loss with the negative GPS scores. The results are listed below:
>
> | Samples          | 5000 | 2500 | 1000 | 500  | 250  |   Ave.$\pm$Std   |
>
> | PREDATOR    | 89.0  | 89.9  | 90.6  | 88.5 | 86.6 | 88.92$\pm$1.53 |
>
> | P + GPS(1, 1) | 89.3  | 89.6  | 90.0  | 88.2 | 88.7 | 89.16$\pm$0.72 |
>
> | P + GPS(1, 2) | 88.6  | 90.2  | 90.3  | 89.2 | 88.4 | 89.34$\pm$0.88 |
>
> | P + GPS(2, 2) | 89.4  | 89.4  | 90.8  | 89.3 | 89.3 | 89.64$\pm$0.65 |
>
> * **(b) Image Matching with LightGlue (Lindenberger et al. CVPR, 2023):** Replacing distances in the loss for SuperGlue (Sarlin et al. CVPR, 2020) with our negative GPS scores is not straightforward, because there are some numerical stability issues in the log-space. Instead, we utilized LightGlue (https://github.com/cvg/LightGlue), which is inherited from SuperGlue, for image matching. Same as we did for PREDATOR, we replaced the feature distances with our negative GPS scores for computing the loss in LightGlue. The results are listed below:
>
> | AUC | 5 degree | 10 degree | 20 degree | Ave.$\pm$Std |
>
> | LightGlue | 41.5 | 57.8 | 70.7 | 56.67$\pm$14.63 |
>
> | L + GPS(1, 1) | 42.5 | 48.3 | 71.2 | 57.33$\pm$14.37 |
>
> As we can see, in both applications, our GPS can significantly improve the baselines in terms of accuracy and robustness. Note that we have not fine-tuned our hyperparameters for GPS in the tables yet. In summary, our GPS can be applied to a wide range of applications for performance improvement.

---

### Meta-Review · Area_Chair_RxTi · 2024-12-20

**Metareview:**

The paper proposes Gumbel Prior Similarity (GPS), a novel probabilistic similarity metric for set-to-set matching based on Gumbel distributions, which model the extrema of distance distributions. It demonstrates that GPS is more robust to outliers than Chamfer Distance while maintaining similar computational efficiency. The proposed method outperforms existing metrics in few-shot image classification and 3D point cloud completion.

The strengths of the proposed method is its computational efficiency and robustness to outliers over the existing methods. The experiments are extensive and cover diverse applications, and the code is submitted for reproducibility.

The major weaknesses are that the paper lacks a deeper theoretical justification for using Gumbel distributions over more general models like the Generalized Extreme Value (GEV) distribution, and it does not address the limitations of its assumptions, such as independence and identical distribution in practical scenarios. It would be better to show the experiments on more practical set-to-set matching problems like image and point cloud matching.

The novelty and technical contribution of the proposed GPS algorithm looks significant, and its simplicity and applicability to various set-to-set matching problems is beneficial to the community.

**Additional Comments On Reviewer Discussion:**

Reviewer 45Az asked for a justification of the use of the Grumbel distribution over other probabilistic models, and a deep discussion on this topic was held. The authors also provided a couple of experiments on the image and point cloud matching problem during the rebuttal, but the results were not reflected in the paper.
After the rebuttal and discussion with the authors, all reviewers agreed to recommend acceptance of the paper, and the AC agrees with their evaluation.

---

### Decision · Program_Chairs · 2025-01-22

Accept (Poster)